# Bytes Are All You Need: Transformers Operating Directly On File Bytes

**Maxwell Horton**                                                                  *mchorton@apple.com*
**Sachin Mehta**
*Apple*

**Ali Farhadi**\*
*Allen Institute for Artificial Intelligence*

**Mohammad Rastegari**\*
*Meta AI*

**Reviewed on OpenReview:** *https://openreview.net/forum?id=RkaqxxAOfN*

## Abstract

Modern deep learning approaches usually utilize modality-specific processing. For example, the most common deep learning approach to image classification involves decoding image file bytes into an RGB tensor which is passed into a neural network. Instead, we investigate *modality-independent* representation learning by performing classification directly on file bytes, without the need for decoding files at inference time. This enables models to operate on various modalities without any hand-designed, modality-specific processing. Our model, *ByteFormer*, improves ImageNet Top-1 classification accuracy by 5% (from 72.2% to 77.33%) relative to DeIT models of similar size. Compared to Perceiver IO, our model requires absolutely no modality-specific processing at inference time, and uses an order of magnitude fewer parameters at equivalent accuracy on ImageNet. We demonstrate that the same ByteFormer architecture can perform audio classification without modifications or modality-specific preprocessing. We achieve 95.42% classification accuracy on the Speech Commands V2 dataset (comparable to the state-of-the-art accuracy of 98.7%). Additionally, we demonstrate that ByteFormer can operate jointly on images and audio, handling joint classification without explicit knowledge of the input modality. We release our code at https://github.com/apple/corenet/tree/main/projects/byteformer.

## 1 Introduction

Deep learning inference usually involves modality-specific processing. For example, Vision Transformers (ViTs; (Dosovitskiy et al., 2020)) explicitly model the 2D spatial structure of images by encoding image patches into vectors. Similarly, audio inference often involves computing spectral features (such as MFCCs (Lyons, 2009)) to pass into a network (Gong et al., 2021; Kim et al., 2021). When a user wants to perform inference on a file stored on disk (e.g. a JPEG image file or an MP3 audio file), the user must first decode the file into a modality-specific representation. Fig. 1a depicts this process for images.

The practice of decoding inputs into a modality-specific representation requires hand-crafting an input representation and a model stem for each input modality. Recent works such as Perceiver IO (Jaegle et al., 2021a) and UnifiedIO (Lu et al., 2022) have shown that Transformer (Vaswani et al., 2017) backbones can be used for a variety of different tasks. However, these methods still require modality-specific input

---

\*Work done while employed at Apple.

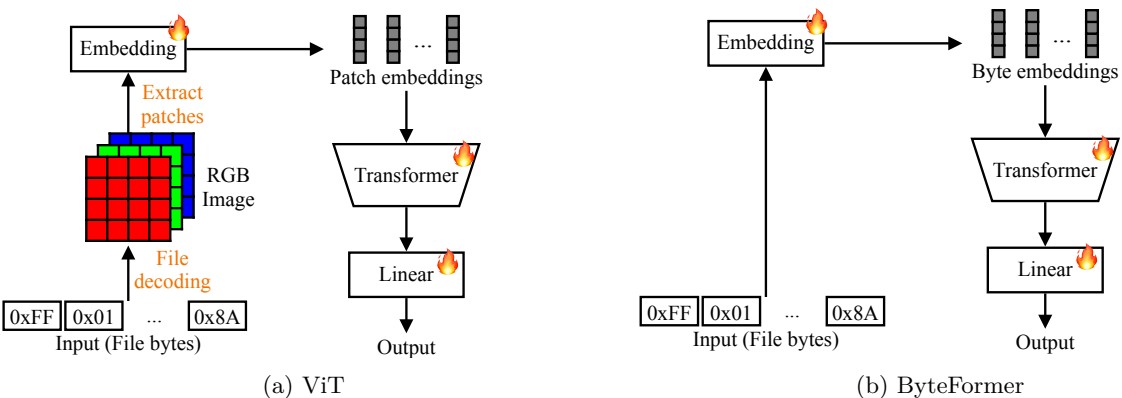

Figure 1: **ByteFormer vs. ViT**. **(a)** A standard vision Transformer (ViT) decodes file bytes into an RGB image. Subsequently, the image is split into patches and patch embeddings are extracted and fed to Transformer to obtain contextualized patch embeddings, which are then classified using a linear classifier. **(b)** ByteFormer directly operations on file bytes.

| Model | Data format | $\mathbb{E}[S]$ | $\mathbb{E}[L_t]$ | Top-1 |
|---|---|---|---|---|
| ViT | RGB Tensor | $3 \times 224 \times 224$ | 196 | 72.20 |
| ViT$^\star$ | RGB Tensor | $3 \times 224 \times 224$ | 196 | **74.35** |
| | fHWC | 150528 | 9407 | **77.06** |
| | fCHW | 150528 | 9407 | 74.65 |
| BF-Ti (Ours) | TIFF | 150668 | 9415 | 77.33 |
| | PNG | 150864 | 9428 | 74.94 |
| | JPEG | 48564 | 12140 | 65.92 |

(a)

| Model | M | Top-1 | Sec | P (M) | F (B) | Im/s |
|---|---|---|---|---|---|---|
| Perceiver | ✓ | 67.60 | - | 55.9 | 62.3 | - |
| Perceiver IO | ✓ | 72.70 | - | 62.3 | 407 | - |
| BF-Ti | ✓ | **77.27** | 1314 | 8.8 | 23.74 | 373 |

(b)

Table 1: **(a)** ImageNet Top-1 accuracy of ByteFormer Tiny (BF-Ti) using various file encodings, compared to ViT. $\mathbb{E}[S]$ denotes the input shape, and $\mathbb{E}[L_t]$ denotes the token length passed to the Transformer backbone. ($^\star$) denotes our implementation of ViT. **(b)** Comparison of ImageNet Top-1 accuracy with Perceiver. **M**: whether the model accepts various modalities (✓: Yes, but with modality-specific modeling. ✓: Yes). **Sec**: Train epoch time (not reported for Perceiver to avoid hardware differences impacting results). **P (M)**: Number of parameters (millions). **F (B)**: Number of flops (billions). **Im/s**: Throughput (images/sec) on an A100 80GB Nvidia GPU. "-" means "not reported".

preprocessing. For instance, Perceiver IO decodes image files and reshapes them before passing them into the network. Other input modalities are processed into different forms.

We hypothesize that it's possible to remove all modality-specific input preprocessing by performing inference directly on file bytes. To test this hypothesis, we develop a Transformer architecture able to operate directly on file bytes. One of the main challenges in operating directly on file bytes is the long token lengths involved. For example, an uncompressed $224 \times 224$ TIFF image stored on disk requires 150668 bytes. To flexibly handle long sequence lengths, we replace the multi-head attention in the Transformer with shifted window attention (Liu et al., 2021), we add token downsampling layers, and we add convolutional downsampling. We call our model ByteFormer (Fig. 1b).

We demonstrate the efficacy of ByteFormer on ImageNet (Deng et al., 2009) classification, achieving 77.33% accuracy on files stored in the TIFF format (Tab. 1a). Our model's backbone is comparable to ViT-Ti (Touvron et al., 2020; Dosovitskiy et al., 2020) (which achieves 72.2% accuracy on RGB inputs). Compared to Perceiver IO (Jaegle et al., 2021a), Our method achieves higher accuracy (77.27% vs 72.70%) with an order of magnitude fewer parameters (8.8 million vs 62.3 million; Tab. 1b). Our model requires absolutely no model-specific processing at inference time (though we do use modality-specific data augmentation during

training, see Sec. 4.3). We also present results on PNG files. Surprisingly, our method is even able to operate on JPEG files, which include complicated compressions like Huffman Codes that aren't byte-aligned.

We demonstrate that our classification model can achieve 95.8% accuracy on Speech Commands V2 (Warden, 2018), comparable to state-of-the-art (98.7%) (Kim et al., 2021), *without any architecture changes or hyperparameter tuning.* We use the same training configuration as for ImageNet, demonstrating the generality of our method. Additionally, a single model can be trained to perform classification of both images and audio, without architecture changes or hyperparameter tuning.

We also investigate multiple file encodings in our image classification and audio classification tasks. Note that using file bytes as our input representation means that our model's performance is dependent on the encoding scheme used. Some encoding schemes more naturally represent data from a given domain than other encoding schemes. For instance, uncompressed TIFF images directly store RGB values, whereas compressed formats such as JPEG store bytes without an easily interpreted semantic meaning. For this reason, we investigate multiple file encodings in our image classification and audio classification tasks to better understand how the chosen encoding scheme impacts performance.

Finally, we perform analyses to understand what patterns ByteFormer learns to recognize. We analyze the learned token embeddings produced by our training procedure, finding that neighboring byte values exhibit strong similarity in uncompressed encodings. From analyzing positional embeddings, we observe that headers are weakly correlated with the image contents for most encodings. We also study the impact of byte ordering on our model's accuracy, finding that encodings that maintain locality produce a higher accuracy.

In summary, our contributions are: (1) To the best of our knowledge, we are the first to explore models that directly consume file bytes without requiring modality-specific processing at inference time. We call our model ByteFormer. (2) We show that ByteFormer achieves strong performance on a variety of image and audio file encodings without the need for architectural changes or hyperparameter tuning. (3) We analyze the embeddings learned by ByteFormer and study the impact of byte ordering on ByteFormer's accuracy. (4) We release our code at `https://github.com/apple/corenet/tree/main/projects/byteformer`.

## 2 Related Work

To the best of our knowledge, previous works use modality-specific modeling in architecture design. By contrast, ByteFormer does not contain any modality-specific modeling. However, our work draws inspiration from previous works, which we discuss in this section.

**Architectures With Multimodal Inputs:** A few methods have explored the idea of feeding different input modalities into the same network for processing. Perceiver IO demonstrates that a Transformer architecture with cross-attention input can be used for a variety of different tasks. However, Perceiver IO decodes file bytes into modality-specific input before feeding into the model. Other recent works explore using Transformers to process multiple modalities (Yu et al., 2023; Lu et al., 2022; Radford et al., 2021; Liu et al., 2023), but also require modality-specific processing. In contrast, ByteFormer operates on file bytes directly, and does not require modality-specific processing at inference time.

**Efficient Attention Computation:** Recent works have explored efficiently handling long sequence lengths in Transformers. These works are applicable to modeling file bytes directly, since the files we consider can have up to $150,528$ bytes. Due to the $\mathcal{O}(n^2)$ dependency on sequence length of the attention calculation, many previous works have suggested modifications to the attention computation to improve efficiency. We experiment primarily with shifted window attention (Liu et al., 2021; Beltagy et al., 2020), which uses fixed-size windows to compute attention. We also explore bag attention (Mehta et al., 2020; Chen et al., 2022) in our ablation study. Bag attention computes attention hierarchically over windows.

**Alternate Image Input Representations:** Previous works have explored using alternate input representations for images. Gueguen et al. (2018) perform partial JPEG decoding, stopping when Discrete Cosine Transform (Marshall, 2001) coefficients are formed. They modify ResNet (He et al., 2015) to ingest this new representation. In (Park & Johnson, 2023), a similar method is used with Transformers. Our work differs in that we perform no decoding of file bytes at inference time.

**Analyzing File Bytes:** Our method avoids modality-specific preprocessing by directly operating on file bytes. Directly analyzing file bytes is a technique commonly used in *binary analysis*, in which computer programs are analyzed for malware content or security issues. See Heena (2021) for an overview. Our work differs in that our primary application is not analyzing the security of computer programs. Our primary appliation is image and audio classification using a machine learning model.

## 3 Method

We describe our architecture and implementation below. We follow the vision Transformer of Dosovitskiy et al. (2020), with a few modifications to handle long sequence lengths.

### 3.1 ByteFormer

An overview of our model is given in Fig. 1b. As input, our model takes in a sequence of file bytes, each of which can take on one of $2^8$ values. We simply treat file bytes as token inputs. The first step of our model is to use a learned byte embedding $\mathbb{E}^{2^8 \times d}$ to convert file bytes into embeddings of size $d$. This differs from ViT's approach (Fig. 1a), which involves decoding file bytes into an image, converting it into patches, and subsequently generating patch embeddings.

Given the large sequence lengths involved when processing file bytes (which can extend up to $150,528$ in our experiments), we employ a strided 1D convolution after generating byte embeddings. This reduces downstream compute and memory usage by reducing the sequence length. Our intuition for choosing strided convolution is that neighboring file bytes often contain related information.

Next, we add positional embeddings to the resulting embeddings and pass these embeddings to a Transformer. Note that the cost of self-attention in Transformers is quadratic with respect to sequence length. To compensate for long sequence length and allow our model to learn hierarchical representations more easily, we make two changes following Swin Transformer (Liu et al., 2021).

The first change is we replace self-attention in Transformers with shifted window attention. Unlike Swin Transformer, our inputs are only 1-dimensional. Thus, our windowing and shifting operations only occur over one dimension. This makes our attention mechanism similar to sliding window attention (Beltagy et al., 2020), but with shifting added.

The second change we make to our Transformer to allow it to handle long sequence lengths is we add down-sampling layers to halve the sequence length. The resulting contextualized byte embeddings are then averaged and fed to a linear layer to produce logits.

### 3.2 Implementation Details

We follow ViT implementation of Touvron et al. (2020). We set model dimension $d = 192$ and use 12 layers of Transformers for learning contextualized byte embeddings. We adopt strided convolution with a kernel size of $k = 32$, and our stride is $k/2$. We chose these settings as they performed well on TIFF images, and we maintained these settings for other experiments. For JPEG, we perform an ablation and find that reducing the kernel size (and stride) improves performance (Tab. 4), likely due to the shorter sequence length obtained by JPEG images. Downsampling layers in ByteFormer appear after Transformer blocks 0, 1, 3, 5, 7, and 9. Our window size for windowed attention is $w = 128$.

## 4 Experimental Setup

When performing inference with a standard Transformer model, the choice of file encoding is irrelevant. For example, it doesn't matter whether an image is stored as a JPEG or PNG file because images are decoded into an RGB tensor before inference. By contrast, ByteFormer performs inference on file bytes. To illustrate the ability of ByteFormer to perform *modality-independent* representation learning, we perform experiments with two different input modalities (images and audio) and multiple file encodings (TIFF, PNG, and JPEG

for images, and WAV and MP3 for audio). This section provides an overview of these file encodings. Note that file encodings typically contain a large number of optional settings that influence the resulting file bytes. We use default settings provided by `PIL` (Clark, 2015) or `scipy` (Virtanen et al., 2020) software packages unless otherwise stated.

## 4.1 Image File Encodings

**fHWC:** We use "fHWC" as an abbreviation for "flattened tensors in height, width, channel order." It refers to uint8 image bytes stored in HWC order without any file headers. It is not common to store images in this way, since they cannot be decoded without pre-existing knowledge of their height and width. This serves as a baseline that demonstrates the performance of ByteFormer on a rasterized image without file headers.

**fCHW:** This format is similar to fHWC, but images are stored in "CHW" order.

**TIFF:** The TIFF file encoding (Parsons & Rafferty, 1998) allows for many custom configurations. For our experimentation we use the default settings provided by PIL, which do not include compression. This results in a format similar to fHWC, but with the addition of TIFF image headers describing configuration options and the image size. Comparing our results on TIFF images to fHWC results helps assess ByteFormer's ability to ignore irrelevant file headers.

**PNG:** The PNG format (Boutell, 1997) contains headers describing PNG configuration options, followed by rows of image data stored in "IDAT" chunks. Each IDAT chunk contains a byte describing the filtering method used for that row's data. The filtering method applies an offset to the row's data based on neighboring pixel values. Thus, our PNG file contains rows of RGB data, with offsets applied, interrupted by occasional bytes that contain file encoding settings. We do not use the optional `zlib` compression that PNG allows. We expect PNG files to provide more challenges to ByteFormer than TIFF, since image contents are encoded as offsets and interspersed with encoding information.

**JPEG:** JPEG (Wikipedia, 2023) encodes images by applying a series of transformations to compress the image before serialization. The RGB image is converted to YCbCr, then downsampled in the chroma channels, then passed through a Discrete Cosine Transform (Marshall, 2001), then quantized using coefficients determined by the JPEG quality factor. The quality factor determines the level of compression, with 100 denoting no compression due to quantization, and lower values indicating stronger compression. After quantization, the coefficients are encoded via a run-length encoding, followed by a Huffman encoding (Raghunathan, 2017). Note that Huffman codes are not byte-aligned, e.g. they can cross byte boundaries. We expect this to make our modeling task more difficult.

## 4.2 Audio File Encodings

**WAV:** The WAV file encoding (Kabal, 2022) stores audio signals represented as a sequence of amplitudes. We use single-channel (mono) audio files. The most common configuration options are the bit depth and the frequency. The bit depth corresponds to the precision with which amplitude values are stored. We experiment with a variety of bit depths, storing audio with 8-bit unsigned integers, 16-bit integers, 32-bit integers, and 32-bit floating-point values. The frequency corresponds to how often amplitude values are chosen. We use 16 kHz, a standard choice for audio (Warden, 2018).

**MP3:** MP3 (Nilsson, 2000) uses a perceptual compression method that removes portions of audio that are difficult for humans to detect. The remaining portions are recorded in frequency space. An MP3 file contains a series of frames. Each frame contains a header with encoding settings, followed by the encoded signal in frequency space. We use standard settings for MP3 provided by the `pydub` (Robert et al., 2018) software package. We expect MP3 encodings to be more difficult to handle than WAV files due to the compression applied.

### 4.3 Preprocessing

Some file encodings such as TIFF and MP3 are not frequently used in machine learning datasets. To allow for comparisons on a single dataset across a variety of file encodings, we must re-encode files with different file encodings.

At training time, we decode the file (e.g. read the contents into an RGB tensor in the case of images, or read the contents into a 1D tensor in the case of audio), then perform standard training augmentation (e.g. random cropping in the case of images, or temporal augmentation in the case of audio), then save the result in the desired file encoding. We find that standard training augmentation is important for model accuracy. Thus, our *training* method is implicitly dependent on the input modality due to our augmentation.

At *inference* time, we do not need knowledge of the input modality. We only need to ensure that our model inputs use the correct file encoding. For example, for TIFF experiments on ImageNet, we precompute $224 \times 224$ crops of the validation images and save them in the TIFF format. Such preprocessing is only necessary because the ImageNet validation set is not already stored in the desired format. Similarly, for audio classification, we re-encode the audio clips in Speech Commands V2 into the desired format before validation.

## 5 Evaluating ByteFormer

One notable advantage of learning representations using file bytes is the potential for a single model to be applied seamlessly across various input modalities, thereby eliminating the need for modality-specific modeling. However, it's crucial to assess the potential trade-off in accuracy associated with modality-independent learning. In this section, we empirically evaluate the capabilities of ByteFormer.

We begin by evaluating ByteFormer on the ImageNet dataset (Deng et al., 2009), demonstrating that it achieves comparable or superior performance compared to ViT (Sec. 5.1). Additionally, we assess ByteFormer on the Speech Commands dataset (Warden, 2018), showcasing its competitive performance against state-of-the-art methods in audio classification (Sec. 5.2).

Furthermore, we extend unimodal evaluations in both images and audio to a multimodal setting. By training a single classifier capable of classifying both images and audio directly from bytes, we show that the resulting model maintains competitiveness with unimodal counterparts (Sec. 5.3).

### 5.1 Evaluating ByteFormer on ImageNet

**Dataset and training details.** We evaluate ByteFormer on 1000-way classification on ImageNet. Our primary comparison is with ViT (Touvron et al., 2020), since our Transformer backbone's size parameters match it. We refer to this architecture as ViT-Ti to emphasize that the distillation in Touvron et al. (2020) is not used. We refer to our architecture as BF-Ti to highlight the fact that our hyperparameters match the "tiny" variant of the Transformer. In spite of our inclusion of shifted window attention in our architecture, we use ViT as our primary baseline rather than Swin since the smallest Swin Transformer Liu et al. (2021) has over $3\times$ the parameter count of our largest model. See Appendix A for comparisons with Swin Transformer.

We train using CVNets (Mehta et al., 2022). For ImageNet, we use a batch size of 48 on a single machine equipped with 8 NVIDIA A100 GPUs. At training time, we use random resized cropping, random horizontal flipping, RandAugment (Cubuk et al., 2019), and RandomErase (Zhong et al., 2017) before storing the image in the desired file encoding (Sec. 4.3). We train with AdamW (Loshchilov & Hutter, 2017) with weight decay 0.05, and a cosine learning rate schedule that anneals the learning rate from 0.001 to 0.00002, with 7500 warmup iterations.

For ImageNet experiments, we report Top-1 accuracy of models trained with exponential moving average (Cai et al., 2021) of weights with momentum 0.0001, which on average increased accuracy by roughly 0.25%.

| Model | Input | $w$ | $k$ | $\mathbb{E}[S]$ | Top-1 |
|---|---|---|---|---|---|
| BC-ResNet-8 | log Mel | - | - | $40 \times 98$ | **98.70** |
| BF-Ti (Ours) | W-FP32 | 128 | 32 | 64058 | **95.80** |
| | | 128 | 16 | 64058 | 95.51 |
| BF-Ti (Ours) | W-INT32 | 128 | 32 | 64044 | 94.90 |
| | | 128 | 16 | 64044 | 95.27 |
| BF-Ti (Ours) | W-INT16 | 128 | 32 | 32044 | 94.81 |
| | | 128 | 16 | 32044 | 95.51 |
| | | 128 | 8 | 32044 | 95.13 |
| BF-Ti (Ours) | W-UINT8 | 128 | 32 | 16044 | 92.28 |
| | | 128 | 16 | 16044 | 94.39 |
| | | 128 | 8 | 16044 | 94.81 |
| | | 128 | 4 | 16044 | 93.99 |
| BF-Ti (Ours) | MP3 | 128 | 8 | 3465 | 88.39 |
| | | 128 | 4 | 3465 | 88.00 |
| | | 32 | 8 | 3465 | 88.69 |
| | | 32 | 4 | 3465 | 89.19 |

| $q$ | $w$ | $k$ | $\mathbb{E}[S]$ | Top-1 |
|---|---|---|---|---|
| 100 | 128 | 32 | 48564 | 60.86 |
| 100 | 128 | 16 | 48564 | 64.86 |
| 100 | 128 | 8 | 48564 | **65.92** |
| 60 | 128 | 32 | 8436 | 31.80 |
| 60 | 128 | 16 | 8436 | 50.11 |
| 60 | 128 | 8 | 8436 | 56.26 |
| 60 | 128 | 4 | 8436 | 62.52 |
| 60 | 32 | 32 | 8436 | 37.23 |
| 60 | 32 | 16 | 8436 | 50.24 |
| 60 | 32 | 8 | 8436 | 56.74 |
| 60 | 32 | 4 | 8436 | 59.52 |

(a)                                         (b)

Table 2: **(a)** ImageNet Top-1 accuracy for ByteFormer Tiny (BF-Ti) for different JPEG quality factors $q$, window sizes $w$, and convolutional kernel sizes $k$. $\mathbb{E}[S]$ denotes the expected shape of the inputs during validation. **(b)** Results for audio classification with BF-Ti on the Speech Commands V2 dataset. "W-" denotes WAV files with the given bit width. $\mathbb{E}[S]$ denotes the shape of network inputs.

**Effect of image file encodings.**   Tab. 1a summarizes results for a variety of file encodings on the ImageNet dataset. For BF-Ti, we use a window size $w = 128$ and convolution kernel size $k = 32$ for all models except JPEG, for which we find $k = 8$ to perform better. Our method surpasses ViT accuracies for TIFF, PNG, fCHW, and fHWC encodings. We note that fHWC outputferforms fCHW by 2.41%. This indicates that channel-wise locality preserves accuracy better than spatial locality.

Notably, our results for all encodings except JPEG surpass the modality-specific baseline. This is likely a result of using a higher parameter count for our embedding layers, and using a longer token length (which requires more computation). More analysis of runtime characteristics compared to modality-specific models appears in Appendix A. We emphasize that our focus is not to obtain a superior efficiency-accuracy trade-off compared to modality-specific models. Our focus is on analyzing the feasibility of avoiding modality-specific processing by using file bytes as inputs.

We find training on JPEG to be more difficult than other modalities. This is due to the highly nonlinear and variable-length JPEG encoding. Note that, since the Huffman coding scheme used in JPEG is not byte-aligned, a byte value's "meaning" is highly dependent on the neighboring bytes. This is in contrast with other encodings like TIFF, in which a byte value's meaning is independent of neighboring bytes (for example, `0xFF` always corresponds to a bright pixel-channel, regardless of neighboring byte values). When using JPEG, our byte embedding presents a challenge since a particular byte value will always be projected to the same embedding regardless of the value of neighboring bytes. Even so, our model is able to obtain 65.92% accuracy.

**Effect of $k$.**   We investigate the influence of our model's kernel size $k$ on JPEG accuracy in Tab. 2a. We find that reducing $k$ from its default value of 32 increases accuracy. Since JPEG images have a smaller token length than TIFF or PNG, they are likely less compressible. To further explore this, we investigate two settings for JPEG quality factor in Tab. 2a. We find that lower quality factors result in lower token lengths, thus reducing $k$ improves accuracy. We also try reducing $w$, but accuracy does not improve.

| Modality | | Balanced? | Epochs | IN | SC2 |
|:---:|:---:|:---:|:---:|:---:|:---:|
| **Image** | **Audio** | | | | |
| ✓ | | - | 300 | 77.33 | - |
| | ✓ | - | 300 | - | **95.80** |
| ✓ | ✓ | ✗ | 300 | **77.47** | 85.71 |
| ✓ | ✓ | ✓ | 300 | 76.64 | 90.08 |
| ✓ | ✓ | ✓ | 150 | 75.46 | 89.81 |

Table 3: Joint image and audio classification compared to unimodal classification with ByteFormer. **Balanced?**: Whether the SC2 dataset is replicated 33× to achieve a balanced dataset in the multimodal case. **Epochs**: The number of training epochs. **IN**: The top-1 on 1000-way image classification. **SC2**: The top-1 on 12-way audio classification.

**Comparison with existing multimodal methods.**   We compare our work with ViT in Table 1a. Byte-Former can surpass the accuracy of ViT when operating on uncompressed files. Note that ViT uses modality-specific preprocessing, and can only be run on images. . We compare our work with Perceiver IO in Table 1b. Our work requires no modality-specific preprocessing, whereas Perceiver requires pixel buffers to be decoded and reordered. Our method uses an order of magnitude fewer flops and parameters than Perceiver (Jaegle et al., 2021b;a). We present more details of our method's computational efficiency compared to related works in Appendix A. Note that previous works include modality-specific processing, which disadvantages our model in the comparisons.

## 5.2   Evaluating ByteFormer on Speech Commands V2

**Dataset and training details.**   We evaluate ByteFormer on 12-way audio keyword classification (including "background" and "unknown" classes) of 1-second audio clips sampled at 16 khz using the Speech Commands V2 dataset. Similar to Kim et al. (2021), we train our model with MixUp (Zhang et al., 2017), noise augmentation, and time shifting augmentation. Then, we store the audio in the desired format. We use the same training and architecture hyper-parameters as in ImageNet (Sec. 5.1) to demonstrate that our method does not require modality-specific hyperparameter tuning. For Speech Commands V2, we found EMA to sometimes increase and sometimes decrease accuracy, so we omit it. We train our models on 4 NVidia A100 GPU machines.

**Effect of audio file encodings.**   Tab. 2b summarizes results for a variety of file encodings on the Speech Commands V2 dataset. BF-Ti achieves accuracies of up to 95.51% on FP32 WAV files. On MP3 files, accuracy is reduced. We believe the compression used in the MP3 format makes the learning task more difficult. This is analogous to JPEG compression reducing accuracy on image classification.

**Effect of $k$.**   We investigate the influence of convolutional kernel size $k$ on model accuracy in Tab. 2b. In general, the optimal $k$ decreases when the expected number of input tokens decreases. This matches our observations in ImageNet JPEG experiments. For MP3 files, we observed that $k = 32$ resulted in unstable models due to the drastic reduction in token length. For MP3, we additionally experiment with window size $w = 32$, but it does not improve results.

**Comparison with existing methods.**   Results for audio classification are given in Tab. 2b. BF-Ti achieves accuracies of up to 95.51% on WAV files, comparable to the state-of-the-art method BC-ResNet-8 (Kim et al., 2021). Note that BC-ResNet-8 is specifically designed for audio processing. By contrast, we performed no parameter tuning relative to our ImageNet training recipe (besides ablating choices of $w$ and $k$). Our best-performing model has window size $w = 128$ and kernel size $k = 32$.

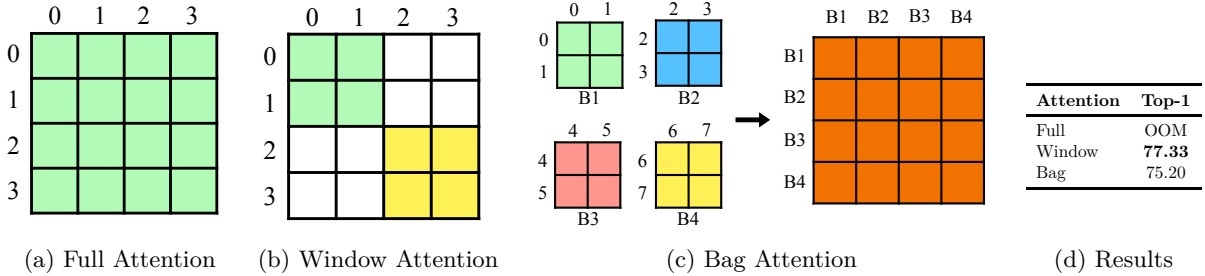

(a) Full Attention     (b) Window Attention     (c) Bag Attention     (d) Results

Figure 2: **(a-c):** Illustration of the types of attention used in ablations. Bag attention is computed in two stages. First, individual bags compute attention. Then, attention is computed across bags. **(d):** ImageNet Top-1 accuracy of BF-Ti with different types of attention. We run out of memory with full attention.

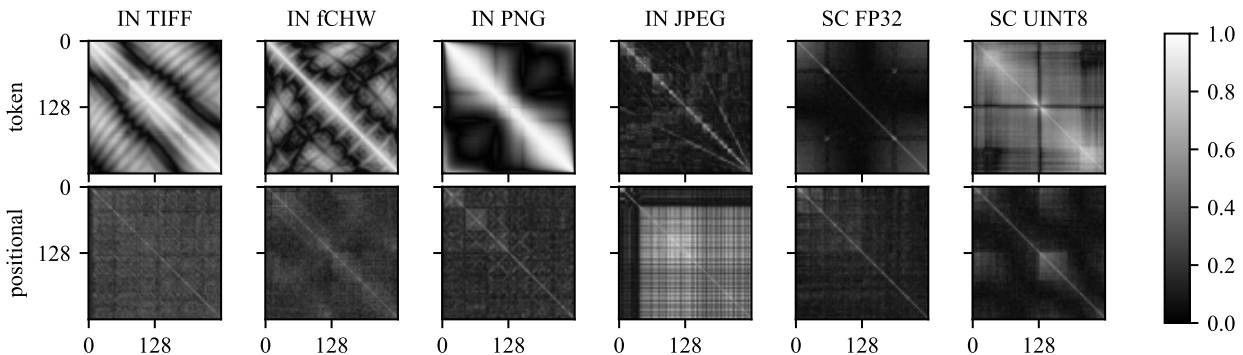

Figure 3: $|x \cdot y|/(||x|| \cdot ||y||)$ for pairs $x, y$ of token embeddings (top row) and positional embeddings (bottom row) learned by BF-Ti. We show results for various file encodings on ImageNet (IN) and Speech Commands V2 (SC).

### 5.3 Evaluating ByteFormer in the Multimodal Setting

**Dataset and training details.** We experiment with training a single classifier to classify both images and audio. To do so, we conjoin the ImageNet and Speech Commands V2 datasets and train a single model for 1012-way classification (since ImageNet has 1000 classes and Speech Commands V2 has 12 classes). We optionally balance the sampling between the two datasets to account for differences in the datasets' numbers of samples by replicating Speech Commands V2 by 33×. Because this doubles the training time relative to ImageNet training, we additionally experiment with reducing the number of epochs from 300 to 150. Note that there is no alignment between the classes represented by these two datasets.

**Comparison with unimodal ByteFormer.** We compare our multimodal models with our unimodal models in Tab. 3. Without dataset balancing, our method produces accuracies on par with the ImageNet TIFF models, but with a lowered accuracy on Speech Commands V2 with WAV-FP32 files. Adding dataset balancing improves SC2 Top-1 results at the expense of a minor reduction in ImageNet accuracy. Note that Speech Commands V2 suffers from overfitting in the balanced dataset case. Accuracy drops from 95.68% (halfway through training) to 90.08% at the end of training.

## 6 Analysis

**Ablating alternate attention methods.** We study three state-of-the-art self-attention methods for handling long sequence lengths in Transformers: (1) full self-attention (Vaswani et al., 2017; Dosovitskiy et al., 2020; Li et al., 2022) where each token attends on every other token, (2) shifted window attention (Liu et al., 2021; Beltagy et al., 2020) where tokens are divided into local windows and each local window computes

| Augmentation | Random Shuffle | Stride | Window Shuffle | Cyclic | Reverse | Baseline |
|---|---|---|---|---|---|---|
| **Top-1** | 3.06 | 5.64 | 18.14 | 60.97 | 61.23 | 60.81 |

Table 4: Ablation showing the Top-1 ImageNet accuracy of BF-Ti on JPEG images using kernel size $k = 32$ and quality factor 100. See text for details.

self-attention independently, and (3) bag (or hierarchical) attention (Mehta et al., 2020; Chen et al., 2022) where tokens are broken up into bags and each bag computes intra-bag self-attention. Inter-bag self-attention is then computed on the resultant output. These different methods are visualized in Fig. 2a while ImageNet results on TIFF inputs are summarized in Fig. 2d. We choose TIFF for these experiments because of its long sequence length (Tab. 1a). We find window attention to outperform bag attention. Note that full attention cannot be run due to its $\mathcal{O}(n^2)$ dependence on sequence length $n$. In our main experiments, we used shifted window attention.

**Learned token embeddings.** We study the token embeddings learned by ByteFormer. These embeddings are used to project file bytes into vector representations. In Fig. 3 (top row), we observe the absolute value of the cosine distance $|x \cdot y|/(||x|| \cdot ||y||)$ between each pair of token embeddings $x, y$ on a variety of file encodings. We choose this metric to highlight the difference between (anti-)correlated embeddings (bright patches) and uncorrelated embeddings (dark patches). The pattern varies substantially across input encodings and tasks. In TIFF, PNG, and fCHW, we observe a bright band off of the diagonal, corresponding to high correlation between bytes and their neighbors. This matches our expectations, since replacing a byte with its neighbor only slightly alters the image. This does not hold for JPEG due to the Huffman encoding step. We also observe that the correlation between token embeddings in the float32 encoding of Speech Commands V2 is generally weak. We believe this occurs because the float32 audio amplitude value is split across four bytes in the file encoding, weakening the association between byte values and amplitudes.

**Learned position embeddings.** We visualize the absolute value of the cosine distance between the first 256 positional embeddings learned by ByteFormer in Fig. 3 (bottom row). For JPEG, we see a strong band of highly uncorrelated values at early positions, corresponding to the file header. Later positions demonstrate interesting patterns that may arise due to the Huffman encodings crossing byte boundaries. In TIFF, a small band of highly uncorrelated values is visible early on, corresponding to the header (which is shorter than in the JPEG case).

**Byte ordering.** To better understand ByteFormer's behavior, we ask, *does ByteFormer simply learn byte frequencies, or is the byte ordering relevant?* In Tab. 4, we apply a series of augmentations during training and validation. We focus on the case of JPEG compression at quality factor 100 with our standard kernel size $k = 32$. Each augmentation modifies the byte order of the inputs in some way. In `random shuffle`, we randomly reorder the bytes during training and validation. The order is redrawn every iteration. This severely degrades accuracy. Next, we perform a strided sampling with stride size 1024 (e.g. $[0, 1024, 2048, \ldots, 1, 1025, 2049, \ldots]$). This slightly improves accuracy over the previous method by improving byte order consistency. Next, we experiment with `window shuffle`, in which the bytes from each window of size 1024 are consistently permuted. This increases accuracy to 18.14%. Next we experiment with a `cyclic` shift in which the second half of the image bytes are moved to the beginning. Accuracy matches the baseline (unaltered JPEG bytes) closely. Similarly, `reverse`, in which the byte order is reversed, preserves locality well and matches the baseline. Our model is sensitive to locality, and does not only learn byte frequencies.

**Inference on obfuscated inputs.** We provide additional experiments that explore the idea of performing inference on obfuscated inputs in Appendix B.

# 7 Limitations

The accuracy of ByteFormer depends on the file encoding chosen. As shown in Sec. 5, choosing JPEG over TIFF results in a reduction of accuracy on ImageNet. Adding invariance to file encodings is future work. Our focus is on analyzing the feasibility of learning directly from file bytes.

Additionally, we note that a separate network is trained for every file encoding in our setup. If inference for a different file encoding is desired, the file will need to be re-encoded in the proper format, or a new network will need to be trained. Given our model's ability to handle multiple modalities in the audiovisual case, we believe a single model could also be trained to handle multiple file encodings. We leave detailed analysis of this question as future work.

Additionally, our method has only been evaluated on classification for images and audio. Experimenting with other domains (video, text) and tasks that require fine-grained localization (detection, segmentation) is exciting future work.

Finally, our method removes modality-specific preprocessing from inference, but still uses modality-specific preprocessing during training in the form of modality-aware data augmentation (Sec. 4.3). Perhaps leveraging modern large-scale datasets would eliminate the need for modality-specific augmentations. We leave this investigation for future work.

## 8 Conclusion

We analyze the feasibility of learning directly from file bytes. To do so, we present ByteFormer, a model that consumes only bytes and does not explicitly model the input modality at inference time. We show that it achieves competitive performance on image classification. The same architecture achieves competitive performance on audio classification without hyperparameter tuning or modality-specific preprocessing. It can also perform multimodal classification of audio and images. In our analysis, we perform ablations of architectural choices used to handle the long sequence lengths inherent in file encodings. We also analyze the learned embeddings and the impact of byte ordering on model accuracy.

### Acknowledgments

We would like to acknowledge useful discussions with Peter Zatloukal, Dmitry Belenko, Mehrdad Farajtabar, Mackenzie Binns, Robert Karl, Minsik Cho, Fartash Faghri, Yanzi Jin, and Mohammad Sekhavat.

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

# A  Performance

Our goal is to directly perform inference on file bytes. To our knowledge, all previous methods involve some level of domain-specific modeling at inference time (including file decoding and different stems for different input domains). Therefore, direct comparison disadvantages our model. Nevertheless, it's important to characterize the runtime of our model, and compare to previous approaches. This helps to contextualize our model's performance.

We compare BF-Ti to related works in Tab. 5. We show performance on ImageNet , including efficiency and accuracy metrics. We only report train time and throughput for models we trained ourselves. This is to avoid hardware differences creating inconsistent results. For these experiments, we tuned BF-Ti's batch sizes to maximize GPU utilization. Note that this improved training time by a relatively small amount (less than 10%) compared to the experiments in Sec. 5.

Our model's size and accuracy (8.82 million parameters, 77.27%) falls between ViT-Ti (5.72 million parameters, 78.62%) and ViT-S (22.05 million parameters, 73.20%). Our model's forward pass time is slower due to the large number of tokens being modeled. Domain-specific modeling (which our model avoids) can drastically reduce compute time.

Compared to Perceiver IO, our model achieves a comparable accuracy with an order of magnitude fewer flops and an order of magnitude fewer parameters. Note that our model performs no domain-specific modeling at inference time. By contrast, Perceiver IO includes domain-specific modeling (file decoding, tensor reshaping, and optionally convolutions) at inference time.

| Model | M | $\mathbb{E}[L_t]$ | **Top-1** | Sec | P (M) | F (B) | Im/s |
|---|---|---|---|---|---|---|---|
| MobileNetv3 Large | ✗ | N/A | 75.10 | - | 5.4 | 0.22 | 9615 |
| ResNet-50 | ✗ | N/A | 78.12 | - | 25.6 | 4.02 | 3488 |
| Swin-T | ✗ | 196 | 81.3 | - | 29 | 4.5 | - |
| ViT-S p16 | ✗ | 196 | 78.62 | 336 | 22.1 | 4.61 | 3594 |
| ViT-Ti p=16 | ✗ | 196 | 73.20 | 334 | 5.7 | 1.26 | 6885 |
| ViT-Ti p=14 | ✗ | 256 | 74.62 | 331 | 5.7 | 1.70 | 4970 |
| ViT-Ti p=8 | ✗ | 784 | 77.44 | 824 | 5.7 | 7.06 | 1243 |
| RGB No More ViT-Ti | ✗ | 196 | 75.10 | - | 5.7 | 1.26 | 6885 |
| Perceiver (learned pos) | ✓ | N/A | 67.60 | - | 55.9 | 62.3 | - |
| Perceiver IO (learned pos) | ✓ | N/A | 72.70 | - | 62.3 | 407 | - |
| Perceiver (conv) | ✓ | N/A | 77.40 | - | 42.1 | 367 | - |
| Perceiver IO (conv) | ✓ | N/A | 82.10 | - | 48.6 | 369 | - |
| BF-Ti k=32 | ✓ | 9415 | 77.27 | 1314 | 8.8 | 23.74 | 373 |
| BF-Ti k=32 -C | ✓ | 9415 | 74.54 | 1122 | 7.6 | 12.63 | 370 |
| BF-Ti k=32 -C -NPE | ✓ | 9415 | 68.42 | 1121 | 5.8 | 12.63 | 372 |
| BF-Ti k=4 f0.05 | ✓ | 3762 | 67.53 | 368 | 6.7 | 5.70 | 1687 |
| BF-Ti k=4 f=0.1 | ✓ | 7524 | 71.26 | 580 | 7.4 | 11.07 | 875 |
| BF-Ti k=8 f=0.25 | ✓ | 9407 | 73.65 | 769 | 7.9 | 15.40 | 634 |

Table 5: ImageNet Top-1 accuracy. **M**: whether the model accepts various modalities (✗: No. ✓: Yes, but with modality-specific modeling. ✓: Yes). $\mathbb{E}[L_t]$: length of token inputs to transformer (after Conv1D for BF-Ti. Note, Perceiver feeds inputs through cross-attention, so this concept doesn't directly apply). **Sec**: Train epoch time. **P (M)**: Number of parameters (millions). **F (B)**: Number of flops (billions). **Im/s**: Throughput (images/sec) on an A100 80GB Nvidia GPU. "-" means "not reported". For ViT, $p$ is patch size. For BF-Ti, $k$ is the convolution's kernel size, and $f$ indicates fraction of retained pixels for masking camera experiments (Appendix B.2). "-C" indicates replacement of Conv1D with a windowed mean. "-NPE" indicates an ablation that removes the positional embedding.

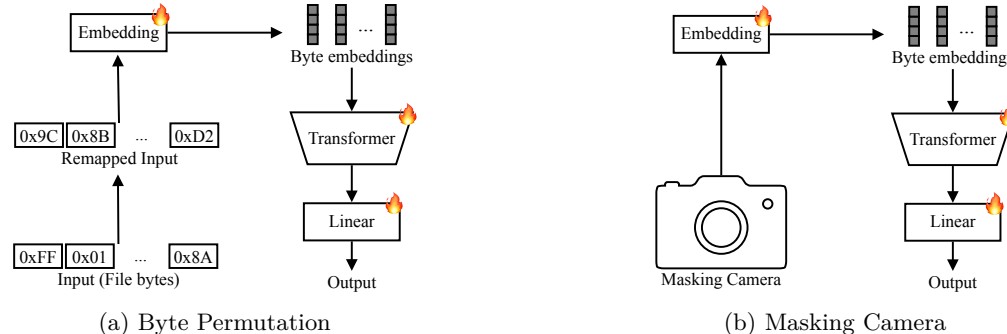

(a) Byte Permutation                    (b) Masking Camera

Figure 4: An overview of our byte remapping method and our masking camera method. (**a**): In our byte remapping method, we remap byte values using a permutation function before passing inputs to our model. (**b**): In our masking camera method, our model inputs are heavily masked images rasterized into a continuous array.

# B    Inference on Obfuscated Inputs

ByteFormer is capable of performing inference directly on file bytes. In this section, we explore the implications of presenting inputs to ByteFormer in non-standard encodings in an effort to obfuscate the inputs.

For example, consider a smart home security system that performs image classification to determine what objects are in the camera's field of view. If an adversary is able to access the inputs to the model due to a security flaw in the system, the adversary can view the contents of the input to the network.

If a model takes in an obfuscated representation of the data, the attack surface that an adversary can leverage could be reduced. There is a natural tension between designing a data representation that supports inference and designing a data representation that preserves privacy. Conceptually, any data representation that supports network inference must leak information about the underlying data, and therefore cannot be fully secure.

In spite of this, obfuscating the inputs to a network can increase the difficulty of an adversary's task, or can reduce the amount of information available to an adversary. In the following subsections, we discuss two ideas that leverage the unique properties of ByteFormer to obfuscate network inputs with minimal loss of accuracy.

We reiterate that these methods alone are not sufficient to provide formal privacy guarantees. We present these ideas to demonstrate that ByteFormer has special properties that can be leveraged to help make an adversary's job more difficult, and to inspire future work in privacy-preserving inference.

In the following subsections, we discuss two ideas for performing inference on obfuscated representations with ByteFormer. In Appendix B.1, we discuss permuting file bytes before passing them to our model (Fig. 4a). In Appendix B.2, we discuss a custom camera capture system that passes masked images to our model (Fig. 4b).

## B.1    Inference on Permuted Inputs

ByteFormer is designed to perform inference on file bytes without converting them into a standard input representation (e.g. an RGB tensor in the case of images). Inspired by this, we demonstrate that ByteFormer can perform inference on images that have been altered in an effort to obfuscate their contents. Specifically, we explore altering the image contents by permuting byte values (Fig. 4a).

We start with an observation about a special property of Transformer models. Consider a permutation $\phi : \{0, 1, 2, \ldots, 255\} \rightarrow \{0, 1, 2, \ldots, 255\}$. Let $\tau$ denote a token embedding, and let $f_\theta$ denote the subsequent Transformer. It's easy to see that, for a given $\phi$, there exists a $\tau_{\phi^{-1}}$ such that $\tau_{\phi^{-1}}(\phi(x)) = \tau(x)$. $\tau_{\phi^{-1}}$ is simply a copy of $\tau$ with embedding vectors reassigned to different indices. Thus, $f(\tau_{\phi^{-1}}(\phi(x))) = f(\tau(x))$.

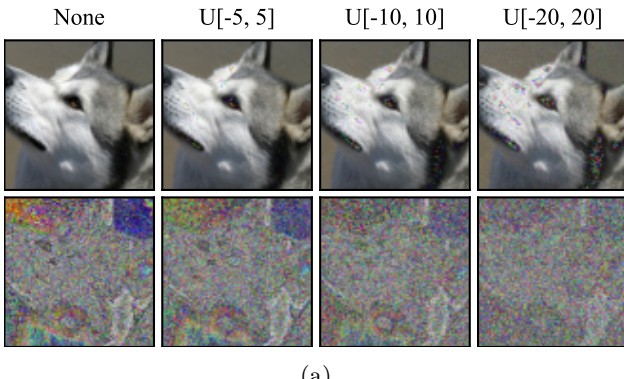

| | None | U[-5, 5] | U[-10, 10] | U[-20, 20] |

| Noise level | Model | |
|---|---|---|
| | **ViT-Ti** | **BF-Ti** |
| None | 51.61 | **77.39** |
| $\mathbb{U}[-5, 5]$ | 50.77 | **77.27** |
| $\mathbb{U}[-10, 10]$ | 49.50 | **77.17** |
| $\mathbb{U}[-20, 20]$ | 43.84 | **76.31** |

(a)                                                                     (b)

Figure 5: **(a):** A sample image from the ImageNet validation set, with uniform noise applied (top row), and with byte remapping $\phi$ additionally applied (bottom row). **(b):** ImageNet Top-1 results for obfuscation with $\phi$. We show results with no noise, and with uniform noise in $[-a, a]$ added. We use the fHWC encoding.

The implication of this statement is that the network $f_\theta$ can operate on re-encoded inputs $\phi(x)$ *without requiring any retraining* as long as the network's token embedding $\tau$ is reordered to $\tau_{\phi^{-1}}$. Since our model operates on token embeddings, this re-encoding method is compatible with our model. By contrast, this re-encoding method is not compatible with the patch embedding used in ViT.

We leverage the above observation to evaluate ByteFormer on obfuscated inputs. Before training, we choose a permutation $\phi$ at random. All training and inference inputs are remapped using $\phi$. More generally, we can use more sophisticated methods for altering input representations by choosing more complicated forms for $\phi$. As ByteFormer can handle highly nonlinear JPEG encodings, we expect it to perform well on a variety of alternative encodings that an outside observer might not be able to easily guess.

Note that the analysis of the effectiveness of this obfuscation against an adversary depends on the threat model used. For example, if an adversary has access to a large number of encoded samples $\phi(x)$, analysis of byte statistics might suggest that strings of common bits correspond to patches of blue sky in images. The adversary's task is made more difficult given certain file encodings (e.g. the highly nonlinear JPEG encoding). We do not make strong claims regarding the level of security provided by different choices of $\phi$.

**Dataset and training details.** We follow the ImageNet training and evaluation setup of Sec. 5.1, using the fHWC file encoding. We add two additional steps to the pipeline.

Our first additional step is, we optionally apply uniform noise to our images. We do this to prevent regions of constant pixel value from appearing in the image, thereby further obfuscating the image. In particular, we add noise from a uniform distribution $\mathbb{U}[-a, a]$ sampled from $-a$ to $a$ (inclusive) to each pixel channel independently, then compute the result modulo 256.

Our second additional step is, we apply a randomly chosen permutation function to byte values before passing them to ByteFormer.

**Results.** Examples of obfuscated images are shown in Fig. 5a. We observe that, without noise, byte remapping retains shape information. A region of the image that is dominated by a single pixel value will continue to be dominated by a new (remapped) pixel value. This reduces the efficacy of obfuscation, since clear patterns are visible in the data. As shown in Fig. 5a, the upper right corner of the image becomes less recognizable as noise from progressively larger ranges is used. In Fig. 5b, we observe that our method is resilient to this obfuscation, but ViT is not. At the strongest level of noise, our model loses only 1.08% accuracy relative to the setting without noise.

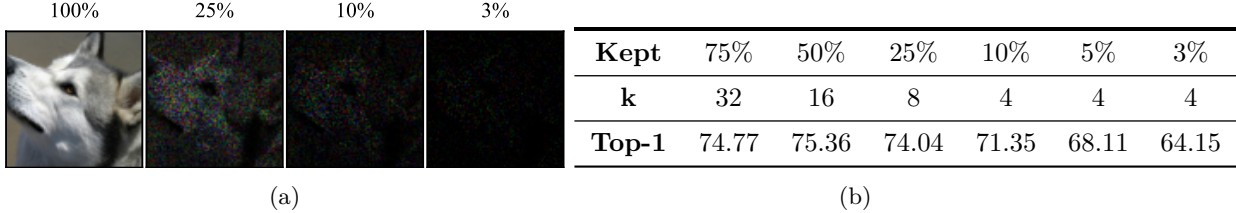

| Kept | 75% | 50% | 25% | 10% | 5% | 3% |
|------|-----|-----|-----|-----|-----|-----|
| **k** | 32 | 16 | 8 | 4 | 4 | 4 |
| **Top-1** | 74.77 | 75.36 | 74.04 | 71.35 | 68.11 | 64.15 |

(a)                   (b)

Figure 6: **(a):** An ImageNet validation image captured by our hypothetical masking camera in which the given fraction of pixel channels are kept. Note that positions of retained pixel channels is discarded by the camera. To make visualization possible, we include the positional information implicitly by placing unmasked pixels in the correct position. **(b):** ImageNet Top-1 accuracy for our masking camera experiment with BF-Ti when the given fraction of pixel channels are kept.

## B.2 Masking Camera

We describe another application of ByteFormer to inference with obfuscated inputs (Fig. 4b). In this scenario, a hypothetical camera captures a non-standard, obfuscated representation to allow for inference without building a full RGB image. This custom representation could take a variety of forms. In our experimentation, we consider a hypothetical "masking camera" that masks out a large fraction of its pixel channels. The camera stores the remaining unmasked pixel channels in an array without retaining the coordinates of pixel channels on the image sensor. In this scenario, an adversary could not obtain a high-fidelity reconstruction of the input image. Even if the adversary could guess pixel channel locations, the adversary could only reconstruct a low-fidelity image due to the high masking ratio used.

**Dataset and training details.** We follow the ImageNet training and evaluation setup of Sec. 5.1, using the fHWC file encoding. To simulate the effect of our hypothetical masking camera, we add one additional step to the pipeline. We precompute a fixed byte mask, and apply the mask to the fCHW bytes. Unmasked bytes are then re-packed into a smaller buffer.

**Results.** Masked images are shown in Fig. 6a. Note that, for ease of visualization, we show images before the unmasked bytes are repacked into a smaller buffer. This means that the visualization in Fig. 6a contains more information than the inputs to ByteFormer, because the byte re-packing destroys spatial information pertaining to where the unmasked pixels are in the original image.

Fig. 6b summarizes our results for our masking camera. For these experiments, we cannot provide ViT-Ti baselines because, unlike ByteFormer, ViT-Ti is not capable of ingesting pixel values without any indication of their placement in the image. This is because of the patch embedding used in ViT-Ti.

At 10% pixel retention, the content of image is hard to visually perceive (Fig. 6a). As shown in Fig. 6b, our accuracy at 10% pixel retention is 71.35%, comparable to the original ViT-Ti model operating on unmasked images.

Note that this masking technique can be combined with the byte remapping technique (Appendix B.1) to further obfuscate network inputs.

