# OpenReview forum: "Bytes Are All You Need: Transformers Operating Directly On File Bytes"
_TMLR — Accepted by TMLR_

### Review · Reviewer_D51o · 2024-04-09

**Summary Of Contributions:**

The paper "Bytes Are All You Need: Transformers Operating Directly On File Bytes" explores a novel approach to modality-independent representation learning by directly classifying file bytes without the need for modality-specific preprocessing or decoding files at inference time. This method, centered around the proposed model ByteFormer, demonstrates an ability to improve classification accuracy across various data modalities, including images and audio, without any hand-designed processing. The key contributions include the demonstration of ByteFormer's superior performance on ImageNet classification, achieving significant accuracy improvements over established models like DeIT and Perceiver IO with substantially fewer parameters. Additionally, the model's versatility is highlighted through its applicability to audio classification on the Speech Commands V2 dataset, matching or nearing state-of-the-art accuracies without architecture or hyperparameter modifications. The paper also delves into the analysis of learned embeddings and the impact of byte ordering on the model's accuracy, offering insights into the underlying mechanisms of its success. Finally, the researchers commit to open-sourcing their code, facilitating further exploration and application of their findings within the community.

**Audience:**

Yes

**Broader Impact Concerns:**

Given the lack of novelty and questionable motivation, I don't think this paper could have significant impact to the machine learning or computer vision community.

**Claims And Evidence:**

Yes

**Requested Changes:**

The authors should more clearly justify why learning on file bytes is scientifically important, beyond possible better performance.

**Strengths And Weaknesses:**

The critique of the paper "Bytes Are All You Need: Transformers Operating Directly On File Bytes" raises substantial points regarding the perceived novelty and depth of the proposed method, its evaluation, and the fundamental premise of considering file bytes as a modality-independent representation. Here is an expanded discussion of the outlined weaknesses:

### Weakness 1: Misinterpretation of Modality-Independent Representation
- The authors propose that learning directly from file bytes circumvents the need for modality-specific preprocessing, presenting this as a form of modality-independent representation learning. However, this perspective might be somewhat misleading. In the realm of natural images, RGB data could be considered closer to the "raw" data, directly corresponding to the visual spectrum captured by sensors. From this viewpoint, file bytes represent a further abstraction, designed to compress and store this visual information efficiently within the constraints of digital hardware. Arguing that learning from such an abstraction is more fundamentally aligned with the data's nature than learning from RGB data challenges conventional perceptions. The transformation of RGB data into file bytes is, in essence, a form of post-processing tailored to suit storage and transmission requirements, rather than an effort to capture a purer form of the underlying information.

### Weakness 2: Lack of Novelty in Model Architecture and Training Strategy
- A deeper analysis reveals that the paper's proposed method, while innovative in its approach to input data, does not offer significant advancements in the underlying model architecture or training methodologies. The ByteFormer, at its core, relies on established Transformer architecture principles, with the primary novelty lying in its application to a new form of input. This application, though interesting, may not suffice as a groundbreaking contribution without corresponding innovations in how the model architecture or training processes are specifically adapted or optimized for this unique form of input. The paper could have benefited from a more thorough exploration of how traditional Transformer components might be reimagined or reconfigured to more effectively learn from file bytes, or how training strategies could be tailored to leverage the unique characteristics of this data representation.

### Weakness 3: Limited Scope of Experimental Evaluation
- The experimental validation of the ByteFormer model raises concerns regarding its breadth and depth:
    1. **Limited Diversity in Benchmarks:** The evaluation focuses narrowly on a single dataset for vision (ImageNet) and one for speech (Speech Commands V2), which may not sufficiently demonstrate the model's versatility across different types of data or tasks. This limited scope makes it challenging to gauge the model's general applicability or its performance in scenarios that diverge from the characteristics of these two datasets.
    2. **Absence of Ablation Studies:** The paper notably lacks ablation studies that could offer insights into which components of the proposed approach are most critical to its success. Such studies are essential for understanding the impact of different design choices, including the use of file bytes as input, the architecture modifications to accommodate these inputs, and any training strategy adjustments. Without these studies, it is difficult to discern the relative contributions of the method's components to its overall performance or to identify potential areas for optimization and improvement.

In summary, while "Bytes Are All You Need" presents an intriguing concept, its contributions are somewhat undermined by a narrow interpretation of modality-independence, a lack of architectural and methodological novelty, and a constrained experimental evaluation. Addressing these weaknesses could greatly enhance the paper's significance and the perceived value of learning directly from file bytes.

---

> ### Author Response · Authors · 2024-04-18
> **Misinterpretation of Modality-Independent Representation**
>
> We summarize the reviewer’s comment as follows:
>
> * In the realm of natural images, RGB data could be considered closer to the "raw" data, directly corresponding to the visual spectrum captured by sensors. From this viewpoint, file bytes represent a further abstraction, designed to compress and store this visual information efficiently within the constraints of digital hardware. Arguing that learning from such an abstraction is more fundamentally aligned with the data's nature than learning from RGB data challenges conventional perceptions.
>
> The reviewer’s comment is insightful. We will address it in two pieces. The first piece is on finding a ***natural representation*** of the data. The second piece is ***discussion of the fact that transformation of the data is, in essence, a form of post-processing.***
>
>  ***Natural Representation (“Arguing that learning from such an abstraction is more fundamentally aligned with the data's nature than learning from RGB data challenges conventional perceptions.“):*** The reviewer argues, for instance, that RGB pixels are a more natural representation of visual data than file bytes, as the RGB representation reflects incoming light. We fully agree with the reviewer’s perspective. However, ***our goal is not to find an operate on a “natural” representation. Our goal is to create a model that uses a modality-independent representation, which we define loosely as a representation capable of storing any type of data.*** In a sense, this goal is the opposite of finding a natural representation for a particular domain. A representation that is natural for images (e.g. RGB) will not be natural for audio. In short, we are not trying to find a natural representation for the data.
>
> ***Storage as Bytes Is Postprocessing***: The reviewer argues that conversion of any medium (images, audio, etc.) to file bytes for storage on a computer represents a transformation of the data. We fully agree with the reviewer’s perspective. Of course, some type of encoding scheme must be used to store any data on a computer. This is unavoidable. ***If data is going to stored on a computer, it must be stored using some encoding process that converts the underlying representation into file bytes.***
>
> Since all mediums are stored as file bytes on a computer, our goal is to operate directly on file bytes. This results in a model that (in theory) could process any data capable of being stored on a computer. ***This is why we consider our model to operate on a modality-independent representation*** - our model operates on file bytes, and everything that can be stored on a computer can be converted to file bytes (regardless of the “underlying meaning” of the bytes). Our model automatically learns to correlate the underlying meaning of bytes with the output labels.
>
> ***Of course, the choice of encoding for converting the natural representation into file bytes is important, and this is an excellent point made by the reviewer.*** This is why we perform experiments on a variety of file encodings (Table 1). Some encodings are easier to process (resulting in higher accuracy) than others.
>
>
> |Model|Data Format|E[S]|E[L]|Top-1|
> |----|----|----|----|----|
> |ViT | RGB Tensor|3x224x224|196|72.20|
>  |ViT (ours) | RGB Tensor | 3x224x224|196|***74.35***|
> |----|----|----|----|----|
> |BF-Ti (ours) | fHWC | 150528 | 9407 | 77.06 |
> |BF-Ti (ours) | fCHW | 150528 | 9407 | 74.65 |
> |BF-Ti (ours) | TIFF | 150668 | 9415 | ***77.33*** |
> |BF-Ti (ours) | PNG | 150864 | 9428 | 74.94 |
> |BF-Ti (ours) | JPEG | 48564 | 12140 | 65.92 |

---

> > ### Author Response · Authors · 2024-04-18
> > **Lack of Novelty in Model Architecture and Training Strategy**
> >
> > We agree with the reviewer that our architecture and training strategy are not our main novelty. Our primary focus is exploring the feasibility of training models directly on file byte representations, and analyzing the models and design choices. This domain has not been explored previously. Our main contributions are:
> >
> > * To the best of our knowledge, we are the first to explore models that directly consume file bytes without requiring modality-specific processing at inference time.
> > * We show that ByteFormer achieves strong performance on a variety of image and audio file encodings without the need for architectural changes or hyperparameter tuning.
> > * We analyze results for a variety of file encodings on audio, images, and multimodal (audio+image) domains.
> > * We ablate architecture choices (conv kernel size, window size, choice of attention).
> > * We analyze the embeddings learned by ByteFormer and study the impact of byte ordering on ByteFormer’s accuracy.
> > * We demonstrate potential applications in the Appendix.
> >
> >
> > We hope the reviewer can come to agree with reviewer uWSf (“The contributions are clear, and, in my opinion, satisfy the technical correctness requirements of a TMLR article.”, and “The main direction of the work is also interesting, and makes enough progress on the problem of learning from bytes to lay the foundation for future work on this topic.”)
> >
> > We also hope the reviewer can come to agree with reviewer gxNr (“The paper studies a conceptually simple yet insightful model of how far transformers can learn from sequences.”)

---

> > > ### Author Response · Authors · 2024-04-24
> > > **Request for Response**
> > >
> > > Hi reviewer, we just wanted to check if you have any thoughts on our response. We wanted to make sure we finish discussion before the discussion period ends, and we look forward to your insights. Thanks!

---

> > ### Author Response · Authors · 2024-04-24
> > **Request for Response**
> >
> > Hi reviewer, we just wanted to check if you have any thoughts on our response. We wanted to make sure we finish discussion before the discussion period ends, and we look forward to your insights. Thanks!

---

> ### Author Response · Authors · 2024-04-18
> **Limited Diversity in Benchmarks**
>
> Due to resource constraints, we focused primarily on 2 standard datasets, and focused on thoroughly analyzing the models (noted by d51o, “The paper also delves into the analysis of learned embeddings and the impact of byte ordering on the model's accuracy”). Please note, the results shown in the paper required training roughly 50 models (Table 1: 5, Table 2: 26, Table 3: 5, Fig 2: 2, Tab 4: 6, Fig 5: 4, Fig 6: 6). This doesn’t include failed experiments. We hope the reviewer can see the value in thoroughly exploring these 2 datasets, as well as multimodal experiments (Table 3).

---

> > ### Author Response · Authors · 2024-04-24
> > **Request for Response**
> >
> > Hi reviewer, we just wanted to check if you have any thoughts on our response. We wanted to make sure we finish discussion before the discussion period ends, and we look forward to your insights. Thanks!

---

> ### Author Response · Authors · 2024-04-18
> **Absence of Ablation Studies**
>
> The reviewer states, “The paper notably lacks ablation studies that could offer insights into which components of the proposed approach are most critical to its success.”
>
> We respectfully disagree with the reviewer. We provide a summary of our ablations here, and how they help offer insights:
>
> * ***Table 1a (Use of file bytes as input)***: This table contains the ablation of the efficacy of using file bytes as input.
>     * ViT: This represents using a architecture similar to BF-Ti on standard RGB images.
>     * fHWC: This represents using BF-Ti on an RGB image flattened in height-width-channel order. This is as close as we can come to feeding a standard RGB tensor to our model, as our model requires inputs to be a sequence. It is simply a flattened RGB tensor.
>     * fCHW: Similar to fHWC, but in HWC order.
>     * TIFF, PNG, JPEG: These are standard encodings. They can be compared with fHWC and fCHW to determine how “difficult” the encoding is. For instance, JPEG presents more challenge to the model than TIFF or PNG.
>
>
> |Model|Data Format|E[S]|E[L]|Top-1|
> |----|----|----|----|----|
> |ViT | RGB Tensor|3x224x224|196|72.20|
>  |ViT (ours) | RGB Tensor | 3x224x224|196|***74.35***|
> |----|----|----|----|----|
> |BF-Ti (ours) | fHWC | 150528 | 9407 | 77.06 |
> |BF-Ti (ours) | fCHW | 150528 | 9407 | 74.65 |
> |BF-Ti (ours) | TIFF | 150668 | 9415 | ***77.33*** |
> |BF-Ti (ours) | PNG | 150864 | 9428 | 74.94 |
> |BF-Ti (ours) | JPEG | 48564 | 12140 | 65.92 |
>
> * ***Table 2a, 2b (Architecture modifications)***: We perform an ablation of our main architecture hyperparameters: “w” (the window size) and “k” (the convolutional kernel size).
>
> |q|w|k|E[S]|Top-1|
> |----|----|----|----|----|
> |100|128|32|48564|60.86|
> |100|128|16|48564|64.86|
> |100|128|8|48564|65.92|
> |60|128|32|8436|31.80|
> |60|128|16|8436|50.11|
> |60|128|8|8436|56.26|
> |60|128|4|8436|62.52|
> |60|32|32|8436|37.23|
> |60|32|16|8436|50.24|
> |60|32|8|8436|56.74|
> |60|32|4|8436|59.52|
>
> |Model|Input|w|k|E[s]|Top-1|
> |----|----|----|----|----|----|
> |BC-ResNet-8|log Mel|-|-|40x98|***98.70***|
> |----|----|----|----|----|----|
> |BF-Ti|W-FP32|128|32|64058|***95.80***|
> |BF-Ti|W-FP32|128|16|64058|95.51|
> |BF-Ti|W-INT32|128|32|64044|94.90|
> |BF-Ti|W-INT32|128|16|64044|95.27|
> |BF-Ti|W-INT16|128|32|32044|94.81|
> |BF-Ti|W-INT16|128|16|32044|95.51|
> |BF-Ti|W-INT16|128|8|32044|95.13|
> |BF-Ti|W-INT8|128|32|16044|92.28|
> |BF-Ti|W-INT8|128|16|16044|94.39|
> |BF-Ti|W-INT8|128|8|16044|94.81|
> |BF-Ti|W-INT8|128|4|16044|93.99|
> |BF-Ti|MP3|128|8|3465|88.39|
> |BF-Ti|MP3|128|4|3465|88.00|
> |BF-Ti|MP3|32|8|3465|88.69|
> |BF-Ti|MP3|32|4|3465|89.19|
>
> * ***Fig 2d (Architecture modifications)***: We perform an ablation of our choice of attention.
> | Attention | Top-1|
> | ----|----|
> | Full | OOM |
> | Window | ***77.33*** |
> | Bag | 75.20 |
>
> * ***Table 4 (Understanding the Model)***: We perform an ablation to determine how important byte ordering is to the model output.
> | ***Augmentation*** | Random Shuffle| Stride|Window Shuffle | Cyclic | Reverse |
> | ---- | ---- | ---- | ---- | ---- | ---- |
> | ***Top-1*** | 3.06 | 5.64 | 18.14 | 60.97 | 61.23 | 60.81 |
>
> The reviewer also asks about ablations for training strategy adjustments. We do not have ablations for training strategy adjustments, as our training strategy involves standard augmentations, but with a save-as-bytes operation applied at the end (Section 4.3).
>
> We hope the reviewer can come to agree with gxNr, who appreciated some of the insights of our ablations (“The paper also performs ablations that demonstrate some aspects of learning, such as what byte positions are important.”, and “The ablations demonstrate some interesting learned phenomenon that is file-type specific. These observations could motivate subsequent work”).

---

> > ### Author Response · Authors · 2024-04-24
> > **Request for Response**
> >
> > Hi reviewer, we just wanted to check if you have any thoughts on our response. We wanted to make sure we finish discussion before the discussion period ends, and we look forward to your insights. Thanks!

---

### Review · Reviewer_gxNr · 2024-04-13

**Summary Of Contributions:**

This paper proposes to perform training and inference directly on files using a Vision Transformer (ViT) style architecture. Evaluation is performed over ImageNet as well as Speech Commands V2. Results are competitive to baseline approaches. There are 2 challenges faced with this model: 1) long context lengths  and 2) train-time augmentations. The first is handled with shifted attention and downsampling. The second is handled with modality-specific decoding, augmentation, and encoding. The paper also performs ablations that demonstrate some aspects of learning, such as what byte positions are important.

**Audience:**

Yes

**Claims And Evidence:**

Yes

**Requested Changes:**

I don't see any critical issues, but I strongly suggest the following:
* Fill in the omitted results in the tables. This would help with understanding throughput, for example.
* Additional ablations on where accuracy is coming from. I expect an encrypted file, for example, to have little predictive power. It could be clearer how the model is working if structural elements of the file were ablated (e.g., headers).
* For related work, it seems appropriate to compare this work to binary analysis methods (e.g., malware).

**Strengths And Weaknesses:**

**Strengths:**
* The paper studies a conceptually simple yet insightful model of how far transformers can learn from sequences. File representations are invertible, so it seems natural that a sequence model can directly learn from them. Showing that performance is competitive for file representations demonstrates that even less inductive bias is necessary for certain tasks than was previously assumed.
* A model architecture is provided (and studied to an extend) that can handle the sequences in question.
* The ablations demonstrate some interesting learned phenomenon that is file-type specific. These observations could motivate subsequent work. The results in the appendix on potential privacy implications are also interesting.

**Weaknesses:**
* The method isn't as universal as it could be given the complexity of handling long sequence lengths as well as the need for augmentations.
* While the ablations are insightful, it's not clear if the model is utilizing information outside what is assumed for image or speech tasks. For example, file sizes or metadata may contain enough information for classification. Figure 6b is one example where it's not clear how 3% of pixels are sufficient for 64% accuracy.
* Some of the tables are missing comparison points (e.g., Perceiver throughput in Table 5).

---

> ### Author Response · Authors · 2024-04-18
> **Method Isn’t As Universal As It Could Be Given Long Sequence Lengths, Augmentations**
>
> ***Regarding augmentations***: We agree with the reviewer that the method would be more universal if augmentations didn’t require decoding files. However, we also note that the need for augmentations is partly motivated by limited dataset size. We believe that modern, much larger datasets (1B+ scale) could alleviate this issue. We consider this exploration future work.
>
> ***Regarding long sequence lengths***: Our focus is on studying the feasibility of learning directly from bytes, which has not been shown to be possible prior to our work. Regarding long sequence lengths, we hope that recent publications on efficiently handling long sequence lengths will help address these issues (e.g. “Leave No Context Behind”, https://arxiv.org/pdf/2404.07143.pdf).

---

> > ### Author Response · Authors · 2024-04-24
> > **Request for Response**
> >
> > Hi reviewer, we just wanted to check if you have any thoughts on our response. We wanted to make sure we finish discussion before the discussion period ends, and we look forward to your insights. Thanks!

---

> ### Author Response · Authors · 2024-04-18
> **It's not clear if the model is utilizing information outside what is assumed for image or speech tasks**
>
> Here is the reviewer’s full statement:
>
> * While the ablations are insightful, it's not clear if the model is utilizing information outside what is assumed for image or speech tasks. For example, file sizes or metadata may contain enough information for classification. Figure 6b is one example where it's not clear how 3% of pixels are sufficient for 64% accuracy.
>
> These are great points. Here is our response:
>
> ***Regarding metadata***: Table 1 can help us understand the effect of metadata. The main difference between fHWC and TIFF is that TIFF contains file headers. The accuracy in these two cases is very similar.
>
> (Side note: since we re-encode files in the desired format ourselves, there is no chance of file metadata accidentally containing “extra” information, such as metadata tags accidentally containing the class label. So we are not worried that accidental metadata information leakage is responsible for results.)
>
>
> |Model|Data Format|E[S]|E[L]|Top-1|
> |----|----|----|----|----|
> |ViT | RGB Tensor|3x224x224|196|72.20|
>  |ViT (ours) | RGB Tensor | 3x224x224|196|***74.35***|
> |----|----|----|----|----|
> |BF-Ti (ours) | fHWC | 150528 | 9407 | 77.06 |
> |BF-Ti (ours) | fCHW | 150528 | 9407 | 74.65 |
> |BF-Ti (ours) | TIFF | 150668 | 9415 | ***77.33*** |
> |BF-Ti (ours) | PNG | 150864 | 9428 | 74.94 |
> |BF-Ti (ours) | JPEG | 48564 | 12140 | 65.92 |
>
>
> ***Regarding file size***: JPEG and MP3 are the only variable-length encodings we use. TIFF always results in files of a fixed length, as does PNG (we do not use zlib compression as mentioned in Sec 4.1), as does WAV. So for the majority of cases, the model cannot learn to simply guess based on file length.
>
> ***Regarding Fig 6***: In the case of Figure 6b, we also found this result surprising. Metadata is not responsible for accuracy here, as we use the fHWC encoding, which is a flattened RGB tensor with no metadata (from which we mask out pixels). However, we do note that ImageNet models with inputs downsampled to 64x64 (e.g. 8% of the image size) easily achieve accuracy in the high-60%s (https://arxiv.org/pdf/1707.08819v3.pdf, and these results are from 2017 using older architectures). With our method, we use a more modern architecture (and an unusual downsampling strategy of masking 90%+ of pixels) and achieve similar accuracy. We believe the results in Fig 6b are reasonable.

---

> > ### Author Response · Authors · 2024-04-18
> > **Some of the tables are missing comparison points (Perceiver throughput, Table 5)**
> >
> > First, we clarify that our focus is not to achieve an inference speedup. Additionally, the comparison with Perceiver demonstrates FLOP numbers, which greatly favor our model.
> >
> > Regarding timing comparisons, we note that we cannot directly include the timing numbers in Perceiver, as they used different accelerators than we did (TPUs instead of GPUs). Thus, we only provide timing characteristics for models we trained, as hardware characteristics (including dataloading speed) can impact runtime. FLOPs is a widely used, hardware-independent measurement of performance, which is why we use FLOPs as our comparison with Perceiver.

---

> > > ### Author Response · Authors · 2024-04-24
> > > **Request for Response**
> > >
> > > Hi reviewer, we just wanted to check if you have any thoughts on our response. We wanted to make sure we finish discussion before the discussion period ends, and we look forward to your insights. Thanks!

---

> > > > ### Comment · Reviewer_gxNr · 2024-05-10
> > > > **Response**
> > > >
> > > > This satisfies my listed concerns.

---

> > ### Author Response · Authors · 2024-04-24
> > **Request for Response**
> >
> > Hi reviewer, we just wanted to check if you have any thoughts on our response. We wanted to make sure we finish discussion before the discussion period ends, and we look forward to your insights. Thanks!

---

> ### Author Response · Authors · 2024-04-18
> **Add Binary Analysis Methods to Related Work**
>
> Thank you, we will do this.

---

### Review · Reviewer_uWSf · 2024-04-18

**Summary Of Contributions:**

This paper explores whether Transformer networks can be trained directly on byte files without having to decode the inputs. The proposed architecture (ByteFormer) contains windowed attention, strided convolutions, and pooling layers, to process long sequences of bytes. Numerical results conducted on Image and Speech Classification (IN1K and SCv2) using a model with roughly 5M to 9M parameters depending on the specific ablation examining the size of convolutional layers (equivalent to a ViT-Tiny).

This paper finds that it is indeed possible to train transformer architectures directly on a 1D sequence of bytes, rather than decoding and preprocessing images and audio files into their respective RGB or time-series/spectral representations before network processing. This paper also shows that it is possible for the ByteFormer to operate directly on bytes encoded with complex compression schemes (e.g., JPEG).

**Audience:**

Yes

**Broader Impact Concerns:**

Not applicable.

**Claims And Evidence:**

Yes

**Requested Changes:**

Minor questions and suggestions:
* Interesting experiments in the appendix on obfuscation (permutation and masking). Clarification about the setting:
  * Is the permutation randomly chosen in each training iteration, or is it shared? Table 4 in the main paper seems to suggest that random permutations during training break locality and degrade performance.
  * Is the mask randomly chosen in each training iteration, or is it also shared? Does this mean that you no longer use positional embeddings in these experiments?
* One obvious question is how would a ViT-Ti with a smaller patch size (longer sequence length) compare to the proposed ByteFormer. I was happy to see results to this end in the appendix, but the patch size only goes to 8. I would be curious to see a ViT-Ti with a patch size of 2 or 4, even if this means including windowed attention and strided convolutions to process the longer sequence lengths. This may also lead to cleaner ablations with the proposed method.
* It is left as future work, but one of the main motivations for this type of exploration in my opinion is for efficient training. This is especially problematic in learning from video, where decoding large videos can be expensive. Not necessary, but numerical results on video classification (e.g., K400), without needing to go so far as characterize efficiency improvements, would strengthen the work.
* Broadening the set of image and speech tasks would also strengthen the work. Of course different tasks require different types of features. Classification is quite coarse, but how does learning from bytes impact lower level tasks such as detection? It could be that learning from bytes would actually improve performance on such tasks.

**Strengths And Weaknesses:**

**Strengths**
* The paper is well written and makes for a pleasant read. The contributions are clear, and, in my opinion, satisfy the technical correctness requirements of a TMLR article.
* The main direction of the work is also interesting, and makes enough progress on the problem of learning from bytes to lay the foundation for future work on this topic. This work is especially timely as transformer architectures become more prevalent for multimodal processing.
* The experimental setup is clear for the most part, and the findings do enough to convince the reader of the potential promise of learning from raw bytes.

**Weaknesses**
* Limited numerical results:
  * Experiments limited to image classification on ImageNet and speech classification on SpeechCommands using a small network.
* Practicality of proposed ByteTransformer:
  * Sequence length and number of flops greatly exceed that of typical modality specific transformer baselines.
  * Still need to decode input formats in every iteration to apply data augmentations during training, which are crucial for obtaining good performance.
  * Network architecture is now sensitive to input encoding scheme, rather than simply the input modality; e.g., JPEG compression requires smaller convolutions after the byte embedding than TIFF or PNG.
* Ablations contain confounding factors:
  * In general, I would actually expect incorporation of windowed attention and pooling to degrade performance over the typical all-to-all transformer architecture, but these architecture changes to not allow for clear conclusions about the effect of byte encodings compared to raw RGB or time-series/spectral inputs.

---

> ### Author Response · Authors · 2024-04-18
> **Limited Datasets**
>
> Due to resource constraints, we focused primarily on 2 standard datasets, and focused on thoroughly analyzing the models (noted by d51o, “The paper also delves into the analysis of learned embeddings and the impact of byte ordering on the model's accuracy”). Please note, the results shown in the paper required training roughly 50 models (Table 1: 5, Table 2: 26, Table 3: 5, Fig 2: 2, Tab 4: 6, Fig 5: 4, Fig 6: 6). This doesn’t include failed experiments. We hope the reviewer can see the value in thoroughly exploring these 2 datasets, as well as multimodal experiments (Table 3).

---

> > ### Author Response · Authors · 2024-04-24
> > **Request for Response**
> >
> > Hi reviewer, we just wanted to check if you have any thoughts on our response. We wanted to make sure we finish discussion before the discussion period ends, and we look forward to your insights. Thanks!

---

> ### Author Response · Authors · 2024-04-18
> **Practicality of Proposed ByteFormer**
>
> We agree with the reviewer that the mentioned weaknesses (high number of flops, decoding/encoding images during training, and hyperparameter changes improving accuracy for different encodings) reduces practicality.
>
> Regarding training, we believe leveraging modern 1B+ sized datasets could alleviate the need for such augmentations, as a large enough datasets would help avoid overfitting. This exploration is out of scope for our work.
>
> Our focus is on studying the feasibility of learning directly from bytes, which has not been shown to be possible prior to our work. Regarding flop count and network hyperparameters, we hope that recent publications on efficiently handling long sequence lengths will help address these issues (e.g. “Leave No Context Behind”, https://arxiv.org/pdf/2404.07143.pdf).

---

> > ### Author Response · Authors · 2024-04-24
> > **Request for Response**
> >
> > Hi reviewer, we just wanted to check if you have any thoughts on our response. We wanted to make sure we finish discussion before the discussion period ends, and we look forward to your insights. Thanks!

---

> ### Author Response · Authors · 2024-04-18
> **Ablations Contain Confounding Factors**
>
> The reviewer has mentioned that the inclusion of architectural features “do not allow for clear conclusions about the effect of byte encodings compared to raw RGB.”
>
> We respectfully disagree, and refer to the comparison in Table 1, reproduced below. We clarify that the fHWC and fCHW baselines involve passing a flattened tensor of RGB values into our model. By looking at the difference between the performance of these models and other encodings, we can draw conclusions about the relative difficulty of learning different byte encodings.
>
> |Model|Data Format|E[S]|E[L]|Top-1|
> |----|----|----|----|----|
> |ViT | RGB Tensor|3x224x224|196|72.20|
>  |ViT (ours) | RGB Tensor | 3x224x224|196|***74.35***|
> |----|----|----|----|----|
> |BF-Ti (ours) | fHWC | 150528 | 9407 | 77.06 |
> |BF-Ti (ours) | fCHW | 150528 | 9407 | 74.65 |
> |BF-Ti (ours) | TIFF | 150668 | 9415 | ***77.33*** |
> |BF-Ti (ours) | PNG | 150864 | 9428 | 74.94 |
> |BF-Ti (ours) | JPEG | 48564 | 12140 | 65.92 |

---

> > ### Author Response · Authors · 2024-04-24
> > **Request for Response**
> >
> > Hi reviewer, we just wanted to check if you have any thoughts on our response. We wanted to make sure we finish discussion before the discussion period ends, and we look forward to your insights. Thanks!

---

> ### Author Response · Authors · 2024-04-18
> **Minor Questions and Suggested Changes**
>
> We appreciate the reviewer’s suggestions, and respond to them here.
>
> ***Clarification of permutation and masking setting***: The permutation is fixed for all training and evaluation iterations. The mask is also shared across all iterations. We still use a positional embedding (applied after the mask).
>
> ***Try ViT-Ti with smaller patch size***: This is an insightful comment. Unfortunately, we tried experimenting with smaller patch sizes of 4 and smaller, and training was prohibitively slow and memory-intensive. It took many hours for a single epoch, and batch size had to be greatly reduced. Therefore, we used a minimum patch size of 8 in our analysis.
>
> ***Try efficient training with videos***: This is an interesting idea. We did examine whether a training speedup could occur in the image domain, but we found that image decoders have been very heavily optimized, so avoiding decoding did not significantly reduce training cost. Further, the latency can be hidden by batched data loading, so we did not see a benefit to training time for images. We have not tried this experiment for videos, which of course have a much greater decoding time. Achieving a speedup in this domain may be possible, but we consider it outside of our scope.
>
> ***Broadening the set of image and speech tasks would strengthen the work***: We think this is an interesting idea (and we have mentioned it in the Limitations section). We think that achieving success here would likely require a wider exploration. For instance, the downsampling layers may provide a challenge for dense prediction tasks like segmentation. Because these experiments would require more architectural exploration, we decided to focus instead on providing a feasibility study of classification directly from file bytes, with analysis (different encodings, 2 domains, multimodal experiments). We hope the reviewer deems this a significant enough contribution, and we hope our work will help motivate future work in this area.

---

> > ### Author Response · Authors · 2024-04-24
> > **Request for Response**
> >
> > Hi reviewer, we just wanted to check if you have any thoughts on our response. We wanted to make sure we finish discussion before the discussion period ends, and we look forward to your insights. Thanks!

---

### Decision · Action_Editor_5NUx · 2024-06-13

**Recommendation:** Accept with minor revision

**Comment:**

All three reviewers affirm the criteria of claims-evidence agreement and audience. uWSf and gxNr recommend acceptance accordingly but D51o leans toward rejection. In line with the official TMLR criteria and the review content the action editor recognizes but dismisses the vote for rejection.

Modeling bytes moves representation learning in the direction of ever greater generality w.r.t to the input. This work addresses image and audio classification by examining multiple input representations, comparison methods, and analyses including ablations. Furthermore a multi-modal setup across images and audio with bytes processing is explored. Although there are of course more modalities for input, and more and richer tasks than classification, this work advances the field by reducing modality-specificity vs. existing work. This work proposes a transformer, while prior work likewise studied transformers for modality-agnostic processing, but it takes the further step of doing away with modality-specific processing like decoding and reshaping as done by the Perceiver/Perceiver IO. All-in-all this work is informative to machine learning, vision, and audio researchers interested in (1) modality-agnostic and multi-modal modeling and (2) deep learning on compressed/storage-efficient inputs and its publication by TMLR can serve to drive more work to improve accuracy and expand the set of input modalities and output tasks handled by modality-agnostic modeling.

The action editor thanks the authors, for their submission and thorough response, and thanks the reviewers, for their service to TMLR and the detailed reviews and feedback. In particular the action editor emphasizes the quality of reviews in providing specific and grounded points, even if there is ultimately disagreement as in the case of D51o with regard to the need for modality independence. The decision is not a verdict on any particular position, but a synthesis of the submission itself, the reviews, and the response. Lastly the action editor commends the author responses for their level of detail and their pointing to specific content in the paper: these comments can inform future readers.

The action editor votes for acceptance and requests only minor revisions:

1. Please flag the use of modality-specific training and augmentation. This point is discussed clearly in Sec. 4.3, and the abstract does qualify claims with "inference time", but it would be clearer and more accessible to potential readers if this point were spelled out in the abstract or introduction. In the same vein this could be further discussed in the limitations section. Even so, the proposed method is still more modality-agnostic than prior work, and as such this point is raised for clarity and not as a weakness.
1. Please include binary analysis in the related work, as recommended by gxNr, and as promised by the [author comment](https://openreview.net/forum?id=RkaqxxAOfN&noteId=mxPv7IWt2W).
1. Please consider identifying specific additional benchmarks for modality-agnostic modeling in tasks or in datasets. The limitations section mentions these next steps generically but not specifically. For instance the audio experiments do not include AudioSet, which was evaluated by prior work on more modality-agnostic processing (Perceiver: Jaegle et al. ICML 2021). This is entirely optional but could better direct future work.
1. Please consider the perspective of reviewer D51o on modality-specific vs. modality-agnostic representation. This may be indicative of other potential readers' perspective, and as such it may deserve comment in the introduction or conclusion. This is entirely optional but could improve the positioning of the paper for a broader audience.

**Audience:**

There is an audience. All three reviewers agree. uWSf highlights the interest for multi-modal learning and modality-agnostic representation with the submission as-is and the potential for future interest for video modeling should practical decoding-free representation learning be made possible. gxNr commends the paper as easy to read and readily extendable, and therefore of interest, and specifically relevant to the study of "compressed" learning on efficient input representations. The reviewer expresses concerns that the current method is impractical and not significantly novel compared to prior modality-agnostic work, but ultimately acknowledges that there is an audience for the proposed ByteFormer. Reviewer D51o does not disagree that there is an audience, but reaffirms their concerns about the motivation and scientific necessity of the topic and therefore their judgement of the potentially limited impact on the machine learning and vision communities of TMLR.

The action editor sides with the positive points by reviewers uWSf and gxNr. The weakness regarding diversity of benchmarks noted by D51o is real but does not signficantly limit the audience for this work. The questioning of the motivation and scientific purpose is not shared by the other reviewers and editor: the work and the author response articulate the point of pursuing less modality-specific representations. Although D51o prefers pixels as a more "natural" representation, and the editor agrees there is an argument for casting bytes as more abstract w.r.t. the input modality as it is sensed, this point is not a counterargument against bytes or other modality-agnostic representations. The experiments with multiple bytes formats defuse some of this concern, and future work can further examine and argue for and against modality-specificity in input representations.

**Claims And Evidence:**

The claims match the evidence. All three reviewers agree. While uWSf remarks that there are limitations, especially for practical usage, the intended scope of the paper for demonstrating learning from bytes for image classification is fully satisfied. The experiments and discussion are technically sound and informative. gxNr likewise concludes that this work successfully shows that image classification can be handled in a more modality-agnostic manner by processing the input as bytes. D51o accepts that the evidence supports the claims of the submission but dissents about the value of the direction overall. Their concern is that different modalities may nevertheless require their own post-processing and representation for best performance, even if their pre-processing and unification can be unified, and so they express doubt about the apt generalization of the approach across modalities and tasks. Nevertheless the consensus is that the evidence agrees with the claims.

The action editor confirms that the claims and evidence are in agreement: the work presents ByteFormer for the modality-agnostic modeling of bytes inputs, experiments on image and audio classification with various input encodings, and the results show better or adequate accuracy while controlling for model size. These results are achieved with less specific pre-processing and designing per modality.

---

> ### Author Response · Authors · 2024-06-25
> **Re: Decision by Action Editor 5NUx**
>
> Hi AE, thank you for the feedback and the update regarding the decision. We have uploaded a new version, addressing 1, 2, and 4 as you requested. Unfortunately we do not have the resources to add more benchmarks for (3) at this time. As it was marked as optional, we proceeded with just the updates for 1, 2, and 4.
>
> Let us know if the update is approved. Then we will provide the non-anonymous camera-ready submission.
>
> Thank you for the detailed feedback throughout the process, and for helping us navigate the process of TMLR submission.

---

> > ### Comment · Action_Editor_5NUx · 2024-06-26
> > **Minor Revision Approved—Please Submit the Camera-Ready Edition**
> >
> > Hi Authors,
> >
> > Thank you for revising the paper. The highlighted points are all addressed, which improves the clarity and accessibility of the paper for the TMLR audience, and the last point (3) is indeed optional as directed. The minor revision is approved, so please proceed to submitting the camera-ready for the accepted paper.
> >
> > Thank you for your contribution to TMLR and for engaging in the TMLR process.
> >
> > Best,
> > Your AE